# Profiling of immune dysfunction in COVID-19 patients allows early prediction of disease progression

André F Rendeiro[1,2], Joseph Casano[3], Charles Kyriakos Vorkas[4], Harjot Singh[4], Ayana Morales[4], Robert A DeSimone[3] (iD), Grant B Ellsworth[4] (iD), Rosemary Soave[4], Shashi N Kapadia[4,5], Kohta Saito[4], Christopher D Brown[4] (iD), JingMei Hsu[6], Christopher Kyriakides[7], Steven Chiu[3], Luca Vincenzo Cappelli[3] (iD), Maria Teresa Cacciapuoti[3], Wayne Tam[3], Lorenzo Galluzzi[2,8,9,10], Paul D Simonson[3], Olivier Elemento[1,2,*], Mirella Salvatore[5,11,*] (iD), Giorgio Inghirami[3,*] (iD)

With a rising incidence of COVID-19–associated morbidity and mortality worldwide, it is critical to elucidate the innate and adaptive immune responses that drive disease severity. We performed longitudinal immune profiling of peripheral blood mononuclear cells from 45 patients and healthy donors. We observed a dynamic immune landscape of innate and adaptive immune cells in disease progression and absolute changes of lymphocyte and myeloid cells in severe versus mild cases or healthy controls. Intubation and death were coupled with selected natural killer cell KIR receptor usage and IgM+ B cells and associated with profound CD4 and CD8 T-cell exhaustion. Pseudo-temporal reconstruction of the hierarchy of disease progression revealed dynamic time changes in the global population recapitulating individual patients and the development of an eight-marker classifier of disease severity. Estimating the effect of clinical progression on the immune response and early assessment of disease progression risks may allow implementation of tailored therapies.

## Introduction

Coronavirus disease-2019 (COVID-19), caused by severe acute respiratory syndrome coronavirus 2 (SARS-CoV-2), is a global pandemic that (as of August 2020) has infected more than 25 million people worldwide, caused more than 840,000 deaths, and strains health systems on an unprecedented scale. COVID-19 has heterogeneous clinical manifestation, ranging from mild symptoms such as cough and low-grade fever to severe conditions including respiratory failure and death (Guan et al, 2020; Richardson et al, 2020). Although most patients with mild disease develop an appropriate immune response that culminates with viral clearance (Guan et al, 2020; Huang et al, 2020; Richardson et al, 2020; Shi et al, 2020 Preprint), severe disease manifestations have been linked to lymphopenia and immune hyperresponsiveness leading to cytokine release syndrome (Guan et al, 2020; Huang et al, 2020; Richardson et al, 2020; Shi et al, 2020 Preprint). The most effective therapeutic approaches developed so far for severe cases involve either general immunosuppression with glucocorticoids (Hennigan & Kavanaugh, 2008) or selective neutralization of IL-6 with tocilizumab (Guaraldi et al, 2020), a monoclonal antibody used to manage cytokine release syndrome in indications such as rheumatoid arthritis (Hennigan & Kavanaugh, 2008). The efficacy of these therapies strongly supports a key role for immune dysregulation in the pathogenesis of COVID-19. However, neither treatment has achieved high clinical remission rates in patients with severe COVID-19 (Kewan et al, 2020; RECOVERY Collaborative Group et al, 2020), suggesting that other immunological or immune-independent attributes may contribute to severity, treatment failure, and ultimately patient death. Thus, in-depth characterization of immune responses to SARS-CoV-2 infection is urgently needed.

Recent characterization efforts have uncovered broad dysregulation of the innate immune system (Schulte-Schrepping et al, 2020b) coupled with altered inflammatory responses (Hadjadj et al, 2020) and impaired adaptive immunity (Zhou et al, 2020). Specifically, the adaptive immune compartment of COVID-19 patients exhibits marked lymphopenia (Huang et al, 2020; Kuri-Cervantes et al, 2020; Mathew et al, 2020), polarization of T cells toward a memory phenotype (Mathew et al, 2020), and functional exhaustion (Blackburn et al, 2009; De Biasi et al, 2020; Zheng et al, 2020a, 2020b),

[1]Institute of Computational Biomedicine, Weill Cornell Medicine, New York, NY, USA   [2]Caryl and Israel Englander Institute for Precision Medicine, Weill Cornell Medicine, New York, NY, USA   [3]Department of Pathology and Laboratory Medicine, Weill Cornell Medicine, New York, NY, USA   [4]Division of Infectious Diseases, Department of Medicine, Weill Cornell Medicine, New York, NY, USA   [5]Department of Population Health Sciences, Weill Cornell Medicine, New York, NY, USA   [6]Division of Hematology/Oncology, Department of Medicine Weill Cornell Medicine, New York, NY, USA   [7]Department of Rehabilitation Medicine at New York University Grossman School of Medicine New York, NY, USA   [8]Department of Radiation Oncology, Weill Cornell Medical College, New York, NY, USA   [9]Sandra and Edward Meyer Cancer Center, New York, NY, USA   [10]Department of Dermatology, Yale School of Medicine, New Haven, CT, USA   [11]Division of Public Health Programs, Department of Medicine, Weill Cornell Medicine, New York, NY, USA

Correspondence: ole2001@med.cornell.edu; mis2053@med.cornell.edu; ggi9001@med.cornell.edu
*Olivier Elemento, Mirella Salvatore, and Giorgio Inghirami are co-senior authors

demonstrating that SARS-CoV-2 infection induces both cellular (Grifoni et al, 2020; Weiskopf et al, 2020) and humoral responses (Schulte-Schrepping et al, 2020b). However, the molecular and cellular mechanisms through which SARS-CoV-2 infection induces these broad immunological derangements in only some patients remains to be elucidated. Furthermore, little is known about the role of the innate immune responses that constitute the first defense against SARS-CoV-2 infection. Moreover, the degree of interaction between various (Alkhouli et al, 2020) immune compartments and demographic factors and medical comorbidities is unclear. The most prominent risk factors for severe disease and death by COVID-19 include age, cardiovascular or oncological comorbidities, and immunosuppression (Yu et al, 2020; Yang et al, 2020a; Guo et al, 2020b). In addition, men appear to be at significantly higher risk for severe COVID-19 than women (Alkhouli et al, 2020). Whereas mortality rates are estimated at 4–6% in the general population, high-risk populations experience mortality rates >60% (Yang et al, 2020b).

Clarifying the early immunological alterations associated with mild versus severe COVID-19 may not only offer therapeutically actionable targets, but also enable the identification of cases at highest risk for clinical deterioration and death. The development of an effective clinical decision-making tool rooted in immunological monitoring has the potential to optimize patient care and resource utilization.

By profiling mild and severe COVID-19 patients and healthy donors with flow cytometry, we demonstrate that SARS-CoV-2 is associated with broad dysregulation of the circulating immune system, characterized by the relative loss of lymphoid cells coupled to expansion of myeloid cells. Severe cases demonstrated enrichment of NK cells expressing the immunosuppressive receptor killer cell immunoglobulin-like receptor, two Ig domains and short cytoplasmic tail 4 (KIR2DS4 and CD158i), and alterations in the B-cell compartment marked by reduced CD19, CD20, and IgM+ cells. These immune profiles enable reconstruction of a hierarchy of disease progression with pseudo-temporal modeling, which allows estimation of dynamic longitudinal changes within individual patients. Our approach also estimates the effect of clinical factors on immune dysregulation and thus establishes an immune-monitoring tool for disease progression.

# Results

## SARS-CoV-2 infection causes major changes in the circulating immune system

We conducted an observational study of 45 individuals with COVID-19 that were treated at New York Presbyterian Hospital and Lower Manhattan Hospitals, Weill Cornell Medicine (IRB 20-03021645) as in- or outpatients between April and July, 2020. The disease was categorized as "mild" if the patient was not admitted or required <6 liters noninvasive supplemental oxygen to maintain $SpO_2$ >92% (n = 21). Patients with "severe" disease required hospitalization and received >6 liters supplemental oxygen or mechanical ventilation (n = 15). Blood samples were collected at enrollment and, when permissible, approximately every 7 d thereafter. Samples were also collected from non-hospitalized individuals who had recovered

from mild, laboratory-confirmed SARS-CoV-2 infection ("convalescent" group, n = 9) and from healthy COVID-19–negative donors (n = 12) (Fig 1A). The median age of COVID-19 patients was 65 yr, which was significantly higher than healthy donors (30 yr) (Tables S1 and S2 and Fig S1).

We performed high-dimensional immune cell profiling of circulating blood by flow cytometry based on seven independent fluorochrome-conjugated antibody panels, each targeting a specific surface protein marker of T, B, NK, and myeloid-derived suppressor cells (MDSCs) (Figs 1B and S2 and Tables S3–S6). Longitudinal sampling was performed in eight patients, one in the "mild" and seven in the "severe" group, and included at least three samples per patient, making a complete dataset including 102 samples from 57 individuals.

Consistent with previous reports (Aschenbrenner et al, 2020 *Preprint*; Hadjadj et al, 2020; Kuri-Cervantes et al, 2020; Mathew et al, 2020), we observed global loss of lymphocytes among CD45+ cells and enrichment of the myeloid cell compartment in the peripheral blood of COVID-19 patients compared with healthy donors (Fig 1C, top). This was exacerbated in patients with severe disease compared with individuals with mild disease (Fig 1C, bottom). This lymphocyte depletion was primarily observed in the T- and NK-cell compartments (Fig 1D). There was no difference in the abundance of B cells between mild and severe groups. These results highlight a major shift in peripheral immune cell absolute abundance from the lymphoid to myeloid lineage (Tables S6).

## SARS-CoV-2 infection causes imbalances in the naive and memory T-cell compartments and induces exhaustion

We next profiled CD4+ and CD8+ T cells in COVID-19 patients and healthy donors (Fig S2). The CD4/CD8 ratio correlated positively with disease severity (Fig 2A) (Mathew et al, 2020; Weiskopf et al, 2020). There was also an expansion of memory T cells (CD45RO+) with reciprocal contraction of the naive compartment (CD45RA+) in severe cases relative to mild disease or healthy donors (Fig 2B). We next quantified the abundance of populations expressing C–C motif chemokine receptor 7 (CCR7; CD197), selectin L (SELL; CD62L), and FAS cell surface death receptor (FAS) (CD95). Within CD45RA+ cells, effector CCR7– ($T_{EFF}$) populations were increased in COVID-19 patients and those with severe disease, especially in the CD8+ compartment (Fig 2C). Conversely, there was significant depletion of CD8+CD45RO+CD95– T cells in patients, which was exacerbated with severe disease.

To characterize these populations more objectively and independently of manual gating, we analyzed the expression of eight surface proteins at the single-cell level by jointly embedding CD3+ cells from all samples with the Uniform Manifold Approximation and Project (UMAP) method and clustering them (Fig 2D). Not all clusters contained cells from all severity groups proportionally. Specifically, clusters 12, 18, and 21, which are characterized by reduced FAS expression, were enriched for cells from healthy donors (Fig 2E). Moreover, there was increased expression of CD95 in samples from COVID-19 patients that correlated with disease severity (Fig 2F and G), and FAS– cells were particularly depleted in all patients (Fig 2G). Indeed, CD95+CD25+ T cells were increased in severe cases, whereas no difference was observed between convalescent patients and healthy donors (Fig 2H).

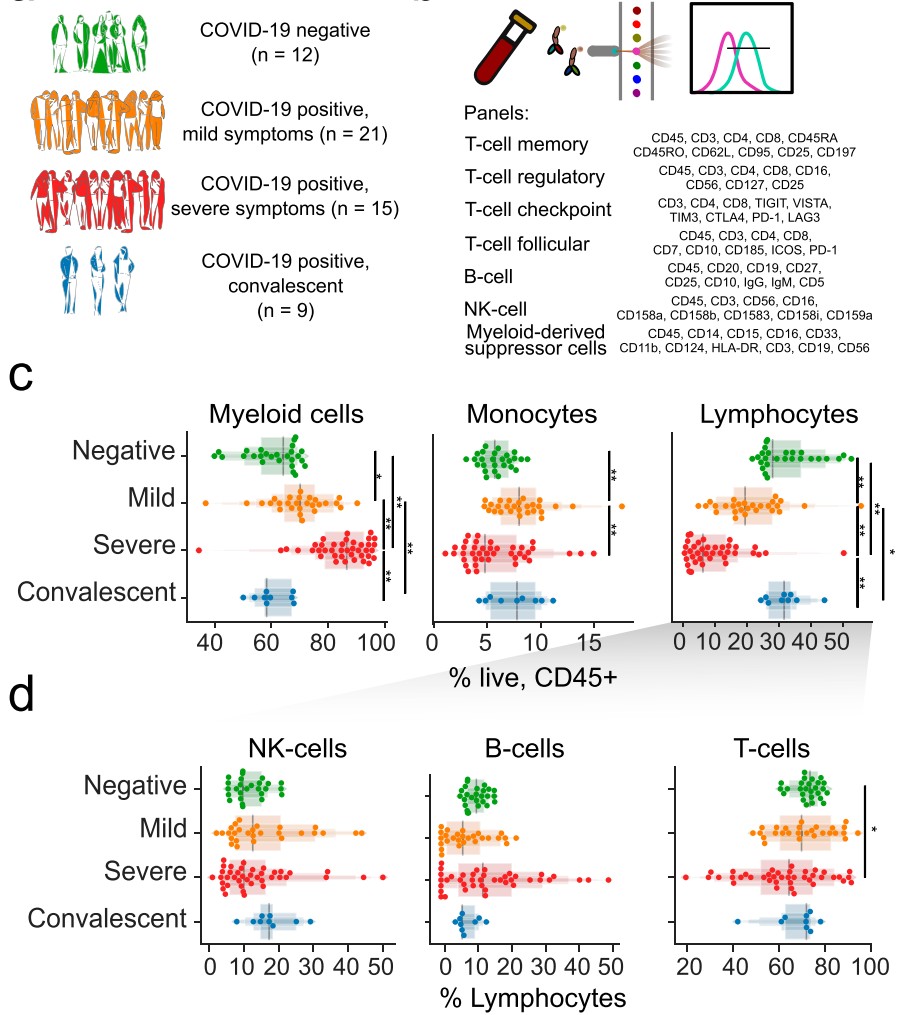

Figure 1. Immunoprofiling of COVID-19 patients reveals a disarrayed immune system.
(A) Composition of the study cohort. (B) Description of immune panels and their target epitopes. (C) Composition of major immune compartments as a percentage of all live CD45+ cells. (D) Abundance of major lymphoid compartments as a percentage of all lymphocytes. For (C) and (D), the upper panels divide patients by general disease status and three lower panels further divide the study subjects by clinical intervention or outcome. Significance was assessed using Mann–Whitney U tests and corrected for multiple testing with the Benjamini–Hochberg false discovery rate (FDR). **FDR-adjusted P-value < 0.01; *FDR-adjusted P-value of 0.01–0.05.

Next, we assessed the frequency of CD4 regulatory T cells (T$_{REG}$, characterized by CD127$^{dim}$CD25$^{bright}$). As markers of follicular helper T cells (T$_{FH}$), we also measured CD4$^+$, CXCR5$^+$, PD1$^+$, ICOS$^+$ T$_{FH}$, which are critical to B cells in the initiation and maintenance of humoral immune responses (Vinuesa et al, 2016). We found a significant but modest increase in T$_{FH}$ in mild and severe COVID-19 cases, with their presence in convalescent patients similar to in healthy donors (Fig S3A and B). However, upon considering a broader spectrum of T$_{FH}$ cells regardless of ICOS expression, CD4$^+$, CXCR5$^+$, and PD-1$^+$ T$_{FH}$ were most abundant in COVID-19 patients with mild disease (Fig 2I). In T$_{REG}$, severe COVID-19 patients showed significant increase compared with healthy donors (Fig S3C), whereas previous reports showed an increase in patients with mild course (Shi et al, 2020 Preprint; Wang et al, 2020).

To investigate T-cell functional phenotypes, we assessed the expression of co-inhibitory T-cell receptors. We observed sustained increase of programmed cell death 1 (PD-1) in COVID-19 patients compared with healthy donors in both CD4 and CD8 compartments. At the same time, V-set immunoregulatory receptor (VISTA) and lymphocyte-activating gene 3 (LAG3) were up-regulated in mild cases (Fig 2J). Exhausted T-cell phenotypes, with high expression of VISTA and LAG3, can be

encountered in chronic viral diseases (Ye et al, 2017), including chronic SARS-CoV-2 infection (De Biasi et al, 2020). This phenotype suggests that these inhibitory receptors may operate at least partially via non-overlapping immunosuppressive signals that negatively regulate T-cell responses during chronic viral infection (Blackburn et al, 2009).

These results highlight a shift toward an activated T-cell memory phenotype in COVID-19 patients, with a potential role for CD95-mediated cell death. By and large, convalescent patients and healthy donors displayed similar immunotypes in comparison with COVID-19 patients. However, we did identify populations such as CD45RA$^+$, CCR7$^+$, CD62L$^-$, FAS$^-$ CD8$^+$ T$_{EFF}$ cells, which remained significantly different to healthy donors up to ~2 mo into recovery (Fig 2C). These cells may represent "T stem memory (T$_{SM}$) cells" with poor expansion potential (Berger et al, 2008) and/or aberrant terminally differentiated effector memory (T$_{EM}$) cells (Laing et al, 2020).

## SARS-CoV-2 induces expansion of polymorphonuclear MDSCs and biases NK KIR usage

Having observed myeloid expansion in COVID-19 patients (Fig 1C), we next investigated the abundance of the MDSC subset. These

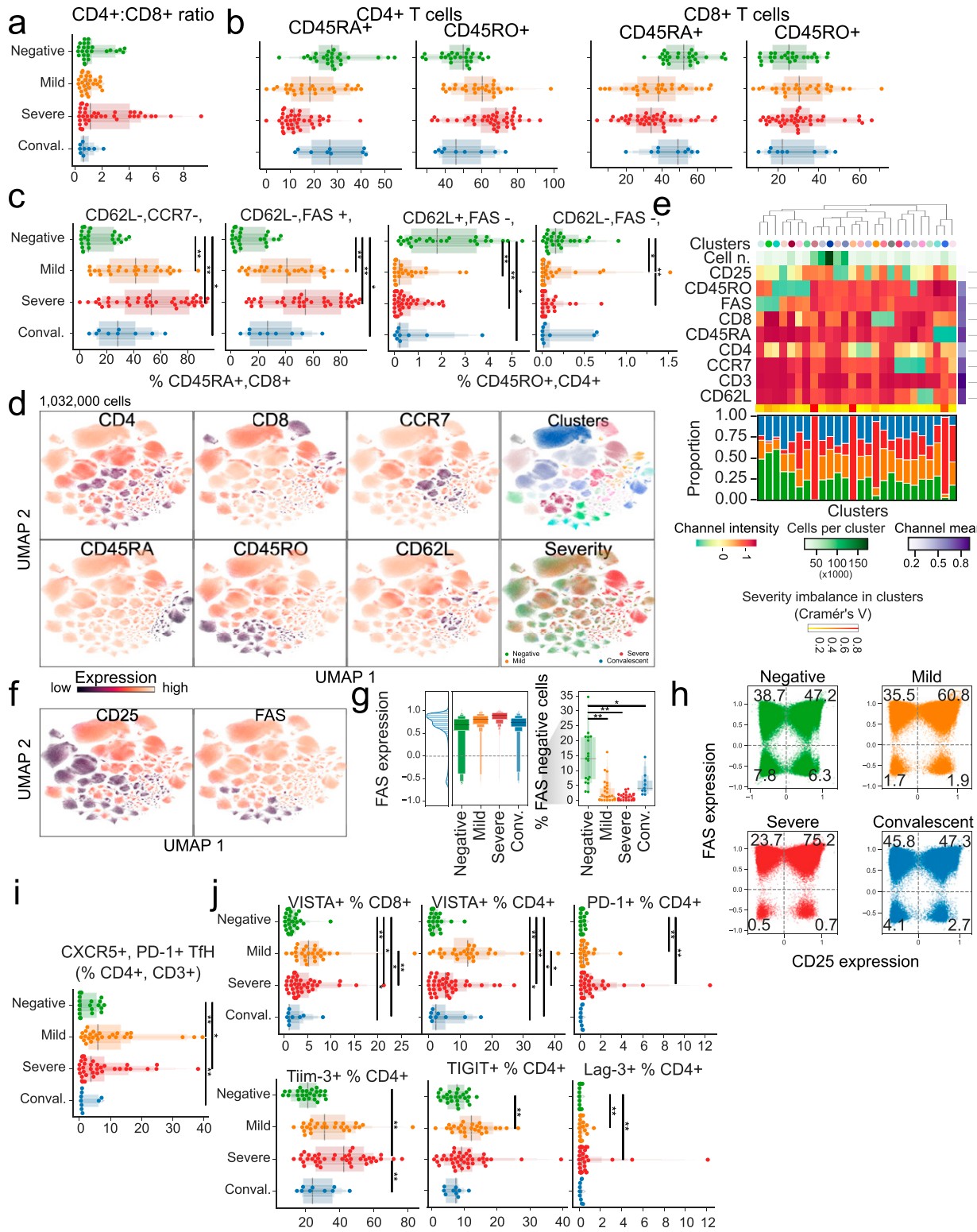

**Figure 2. T cells from COVID-19 patients have high levels of CD25, FAS, and exhaustion markers.**
**(A)** The ratio of CD4 to CD8 cells is dependent on disease state and clinical intervention. **(B)** The abundance of CD45RA/RO cells in either CD4+ or CD8+ compartments is dependent on disease state or clinical intervention. **(C)** Abundance of immune populations changes significantly between disease states. **(D)** Uniform Manifold Approximation and Projection (UMAP) projection of all cells colored by either surface receptor expression, cluster assignment, or disease severity. **(E)** Immune phenotype of each cluster (top) and its composition in disease severity (bottom). **(F)** Expression levels of CD25 and FAS receptors in the UMAP projection. **(G)** FAS expression across all clusters depending on disease severity (left) and the proportion of cells not expressing FAS for each sample (right). **(H)** Scatter plot of CD25 and FAS expression for each

elements are activated by IL-6 (Tobin et al, 2019) and have immunomodulatory functions in cancer (Gabrilovich & Nagaraj, 2009; Kumar et al, 2016) and viral infections (O'Connor et al, 2017). Our flow cytometry panel considered CD3⁻, CD56⁻, CD19⁻, HLA-DR⁻/dim, CD33⁺, CD11b⁺ cells and focused on distinguishing CD14⁻, CD15⁺ granulocytic cells (G-MDSCS); CD14⁺, CD15⁻/dim monocytic-like cells (M-MDSC); and CD14⁻, CD15⁻/dim immature cells (I-MDSC) from each other. G-MDSCs were rarely detected in healthy donors but were prevalent in mild and severe COVID-19 patients (Fig 3A and B). Convalescent patients showed numbers of G- and M-MDSCs closer to healthy donors, with a nonsignificant increase in I-MDSCs compared with healthy donors. Conversely, I-MDSC cells, although relatively rare as a fraction of all immune cells, were further reduced with disease (Fig 3A and B). Because neutrophils are phenotypically similar to MDSCs, we compared their abundance with MDSCs. Although there was a positive correlation between G-MDSCs and a high neutrophil count, neither could account entirely for the other (Fig S4).

Next, we created a joint embedding of 2.4 million CD16⁺ cells from all samples using the UMAP method, deriving clusters based on similar cells (Fig 3C). Clusters containing CD15⁺ cells were disproportionately enriched in samples from COVID-19 patients, whereas clusters with CD3⁺, IL4R (CD124) were mostly composed of cells from healthy donors (Fig 3D). In addition, CD15 expression was most prominent in COVID-19 patients, particularly in severe cases, but when selecting for CD3⁻ or CD3⁻ CD33⁺ cells, convalescent patients possessed a number of CD15⁺ cells more similar to patients with active disease than healthy donors.

Next, we focused on innate lymphoid cells and determined the expression of KIR receptors in CD56⁺, CD16bright NK cells. Whereas we observed no significant differences in the relative abundance of KIR receptors among COVID-19 patients with mild disease and healthy controls (Fig 3E), a significantly higher proportion of cells expressed CD158i (KIR2DS4) in severe patients than in mild or convalescent individuals. Moreover, we observed fewer CD158e (KIR3DL1) cells in patients with mild disease compared with severe patients and a lower proportion of cells not expressing any of the measured receptors (KIR⁻) in patients with severe disease. To further explore NK cell subsets independent of conventional gating, we harnessed single-cell analysis and integrated >500,000 cells in a UMAP representation, identifying cell clusters based on surface marker expression (Fig 3F). Clusters significantly enriched in CD158i-expressing cells were paucicellular in healthy donors compared with COVID-19 patients (Fig 3G), and the relative frequency of CD158i-expressing cells was lower in healthy donors, regardless of the expression of other KIR receptors (Fig 3H). Because the expression of KIR variants is stochastic, the apparent selection of KIR-expressing cells in severe COVID-19 patients could indicate that a viral antigen presented by MHC class I molecules with higher affinity for CD158i could select for NK cells expressing this receptor.

### B cells of COVID-19 patients show distinct patterns of immunoglobulin expression associated with disease severity

Because B cells play a critical role in adaptive immunity, we investigated the expression levels of surface CD19, CD20, IgM, and IgG

in circulating cells. Despite the backdrop of a relative decrease in B cell numbers as disease progresses, we observed only a mild, nonsignificant increase in plasmacytoid cells in patients with severe COVID-19 compared with healthy donors (Fig 4A). However, the number of IgM⁺ CD19⁺ CD20⁺ B cells was decreased in patients with severe disease compared with mild, whereas IgG⁺, CD19⁺, CD20⁺ cells remained comparable across all patients (Fig 4B). Next, we visualized single cells from all patients in a common UMAP plot and assigned clusters based on surface marker expression (Fig 4C). This approach identified two distinct groups based on the expression of surface IgM, with the total number of IgM⁺ cells within clusters increased in severe COVID-19 patients (Fig 4D). Conversely, healthy donors displayed B cells with high expression of surface CD19⁺ and CD20⁺ antigens (Fig 4D). Closer inspection of CD19 and CD20 expression identified two distinct populations that differ in CD20 levels (Fig 4E). This approach also revealed that the relative abundance of circulating CD19 and CD20bright B cells was lower in COVID-19 patients compared with healthy individuals regardless of disease severity.

To shed light on the functional relevance of these different B cell subsets, we quantified the expression of IgG and IgM in each population identified based on CD19 and CD20 co-staining (Fig 4F). Circulating CD19low B cells (populations A and B) were enriched for IgG⁺ cells in patients with mild and severe COVID-19 and IgM⁺ cells in severe COVID-19 patients, whereas convalescent patients resembled healthy donors. No such difference was observed with CD19⁺ and CD20bright B cells (population C) and CD19⁺ CD20⁺ and CD19⁺ CD20⁻ B cells (populations D and E). Overall, despite dwindling numbers of B cells overall, specific subsets of B cells, especially those with lower CD19 expression, have distinct immunoglobulin expression patterns in COVID-19 patients, with severe patients more frequently bearing IgM⁺ B cells. We speculate that these findings may be related to the plasmacytoid differentiation and immunoglobulin switching programs, which may be dysfunctional due to SARS-CoV-2 infection.

### Pseudo-temporal modeling unveils a highly dynamic immune cell landscape of COVID-19 over time

Having characterized the main circulating compartments of the immune system, we next sought to leverage the high dimensionality of the dataset and hypothesized that its underlying data structure would be useful for reconstructing the clinical course of COVID-19. Thus, we used pseudotime inference to reconstruct an underlying latent space from a healthy state to a severe disease state (Fig 5A and B).

Further analysis of the inferred space enabled identification of circulating immune cell populations associated with disease progression. In particular, we identified a space driven by a decrease of lymphocytes, gain of myeloid cells (G-MDSCs in particular), and a terminally activated/exhausted T cell phenotype (Fig 5C). Besides discovering immune signatures associated with each degree of severity, this analysis allows the relative positioning of

cell according to disease severity. **(I)** Abundance of CD4⁺ CXCR5⁺ PD-1⁺ T_FH by disease severity. **(J)** Immune populations with significantly different amounts of cells expressing immune checkpoint receptors by disease severity. Significance was assessed by Mann–Whitney U tests and corrected for multiple testing with the Benjamini–Hochberg false discovery rate (FDR). **FDR-adjusted *P*-value < 0.01; *FDR-adjusted *P*-value 0.01–0.05.

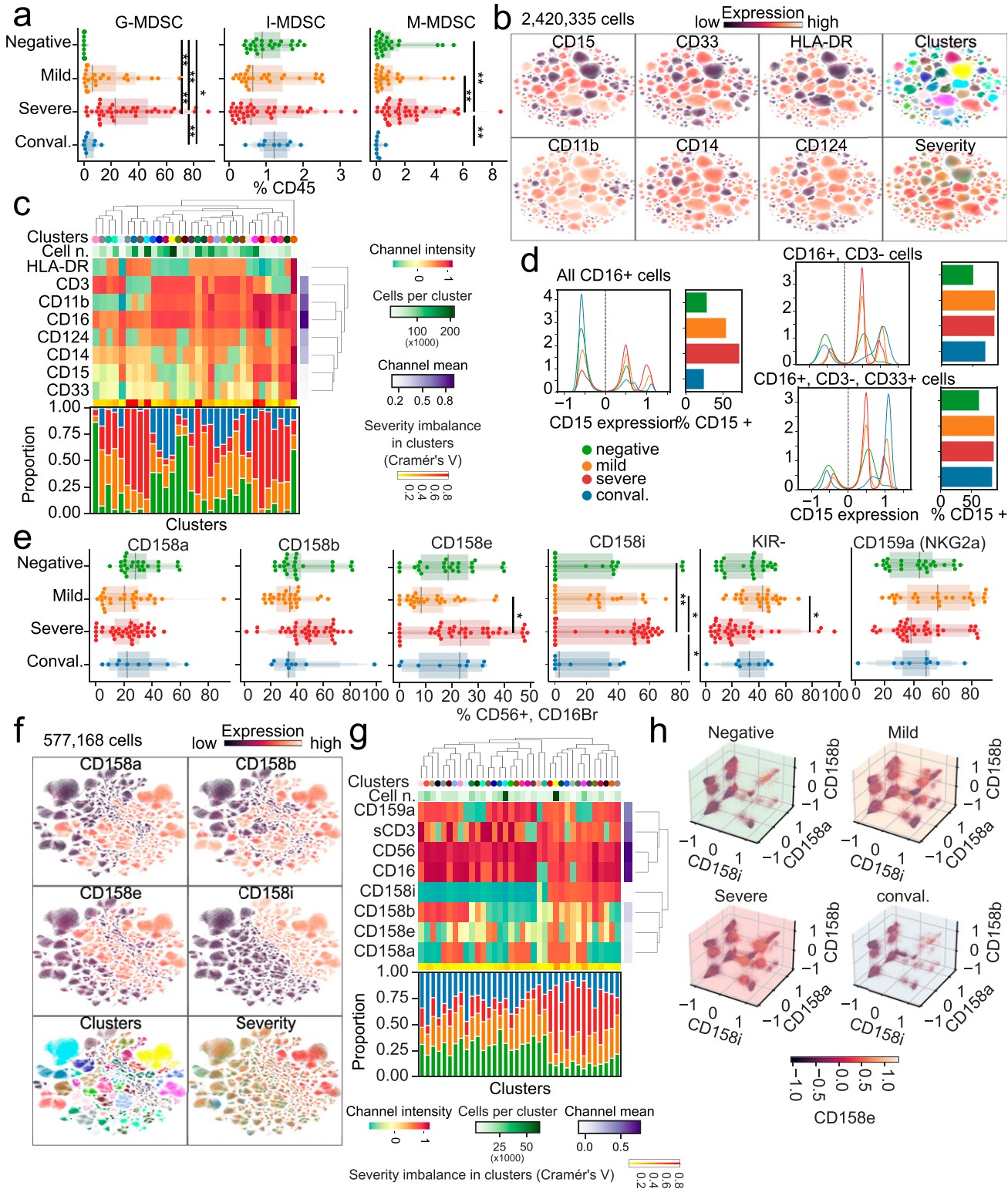

**Figure 3. Emergence of granulocytic myeloid-derived suppressor cells and preferential expression of specific NK cell receptors in the innate immune system of COVID-19 patients.**

**(A)** Abundance of myeloid-derived suppressor cells as a percentage of all immune cells according to disease severity. **(B)** Uniform Manifold Approximation and Projection (UMAP) projection of all cells from all patients colored by the expression levels of surface receptors, derived clusters, or disease severity among all patients. **(B, C)** Immune profile of each cluster from (B) based on the expression of surface markers (top) and composition in disease severity (bottom). **(D)** Expression levels of CD15 dependent on disease severity (left) and quantification of cells expressing it (right) according to CD16, CD3, and CD33 expression. **(E)** Abundance of cells expressing various KIR

each time point in relation to the continuous changes characterized by pseudotime (Fig 5D). Variable changes associated with the pseudo-temporal axis could be classified in three clusters (Fig 5D). The first was composed of 68 populations, with an increase toward higher disease severity with representatives such as the fraction of myeloid cells, PD-1$^+$ CD4$^+$ T cells, and CD62L$^-$ cells among CD45RA$^+$, CD8$^+$ T cells (Fig 5E left). The second corresponded to a virtually stable cluster with 52 populations such as IgM$^+$ B cells, with only mild fluctuation in the intermediate stage (Fig 5E, center). The third included a cluster with a steady decrease by disease severity, encompassing the overall lymphoid population as well as B cells and CD45RA$^+$ CD4$^+$ T cells (Fig 5E, right). This effectively establishes a temporal hierarchy of changes as disease progresses in which populations such as CD62L$^+$, CCR7$^+$, CD45RA$^+$, CD8$^+$ T cells have a steady decline and others such as B cells have a stronger decline toward the severe end of the pseudo-temporal timeline. In addition, the dynamic character of changes raises the possibility of using flow cytometry to improve COVID-19 patient stratification based on real-time immunological monitoring. Although our observations do not indicate causality, immunological variations in the pseudo-temporal dimension may offer testable hypotheses on COVID-19 progression mechanisms.

### Integration of clinical and demographic factors affecting COVID-19 immunity and stratification of patients by disease severity

Because various clinical and demographic factors influence disease incidence and mortality (Guan et al, 2020; Richardson et al, 2020; Zhao et al, 2020), we investigated the interaction between SARS-CoV-2 infection, the circulating immune system, and various demographic and clinical factors. Thus, we fit regularized linear models to the proportional flow cytometry data with covariates such as sex, race, age, disease severity, presence of comorbidities, hospitalization, intubation, and death (Fig S5A). We also estimated the interaction of sex with clinical variables such as disease severity, hospitalization, intubation, and death. The resulting network of significant effects identified several clinical factors associated with specific immune cell populations, highlighting how age, sex, and disease severity jointly influence the circulating immune systems in patients with COVID-19 (Fig 6A).

As a baseline, we could recover known effects independent of disease, such as a higher CD4:CD8 ratio in females than males and an overall decrease of the lymphoid population with age (Fig S5B). Last, we found associations between sex and clinical variables such as a significantly higher fraction of CD62L$^+$, CCR7$^+$, CD45RO$^+$, CD4$^+$ T cells in males that died compared with females (Fig 6B, left) and much lower total lymphocyte levels in females that died compared with males (Fig 6B, right). Regarding the effect of tocilizumab on the immune system, we compared posttreatment samples from eight treated severe patients to seven severe untreated patients. Although we observed the largest effect in certain subsets of CD4$^+$ T cells, there was also an increased relative abundance of B cells and a decrease in T cells expressing the co-inhibitory receptor hepatitis

A virus cellular receptor 2 (HAVCR2; TIM3) (Fig S5C). Moreover, the signature associated with severe versus mild patients was broadly counteracted by tocilizumab (Fig 6C). Associations between sex and clinical variables were found, such as a lower fraction of CD62L+, CCR7+, CD45RA+, CD8$^+$ T cells in females treated with tocilizumab compared with males, contrary to the opposing trend in untreated individuals (Fig 6D), or the lower frequency of CD158a NK cells in female intubated patients (Fig 6E).

Because there is a need to stratify patients to provide better, more effective, and less costly care, particularly in the earlier stages of disease, we hypothesized that the high dimensionality of the immunotypes would make it possible to train a classifier to predict disease severity early on. A random forest classifier was trained to distinguish patients with mild from severe disease using only the earliest available sample of each patient in a cross-validated manner (Fig 6F). We observed good performance of the classifier (median area under receiving operator curve [ROC AUC], 0.81) compared with one with randomized severity labels (median ROC AUC, 0.49) (Fig 6G), providing good balance between true positive and false positive rates. Because our dataset is composed of immune populations from seven flow cytometry panels, we tested whether a smaller number of variables could discern patients with mild and severe disease courses. With only eight variables, the classifier could distinguish patients with different disease severities, albeit with lower performance (ROC AUC, 0.73 versus 0.49 with randomized labels) (Fig 6H). Furthermore, we hypothesized that our classifier could be used for real-time immunomonitoring of COVID-19 patients. Thus, we applied it to subsequent samples of patients with more than three samples collected over the disease course, while withholding those samples from the training set (Fig 6I). Patient 26, who had an overall mild disease course, had all samples classified as mild; severe patients often showed dynamic severity probabilities over time, with at least one time point classified as severe disease. To exemplify how this prediction relates back to flow cytometry data, we illustrate the aggregated expression of the activation marker CD25 and CD45RA/RO in single T cells over time (Fig 6J). Patients with lower predicted severity toward the end of their course (e.g., patient 23) tended to have less CD25 expression and increased CD45RA expression, whereas the opposite was also true (e.g., patient 16). Patients with predictions that were either more stable or dynamic over time (patients 26 and 24, respectively) showed dynamics of expression in accordance to their overall predicted pattern over time. This proof-of-principle work demonstrates our ability to leverage high-content immune profiling to predict overall disease course and provides the basis for real-time immune-monitoring of COVID-19 patients.

## Discussion

Here, we describe the circulating immune landscape of COVID-19 patients compared with healthy individuals. Consistent with previous reports (Kuri-Cervantes et al, 2020; Mathew et al, 2020; Wang

---

receptors as a percentage of NK cells according to disease severity. **(F)** UMAP projection of all cells from all patients colored by the expression of surface receptors, derived clusters, or disease severity. **(F, G)** Immune profile of each cluster from (F) based on the expression of surface markers (top) and composition in disease severity (bottom). **(H)** Expression levels of all four measured KIR receptors in each disease state. Significance was assessed using Mann–Whitney U tests and corrected for multiple testing with the Benjamini–Hochberg false discovery rate (FDR). **P-value < 0.01; *FDR-adjusted P-value 0.01–0.05.

Figure 4. **B cells of COVID-19 patients are marked by a shift toward a plasmocytic IgM phenotype.**
**(A, B)** The abundance of total B cells, plasma, and IgG+ and IgG+ cells between disease states. **(C)** Uniform Manifold Approximation and Projection (UMAP) projection of all cells colored by surface receptor expression, cluster assignment, or disease severity. **(D)** Immunophenotype of each cluster (top) and its composition by disease severity (bottom). **(E)** Identification and quantification of five populations of B cells dependent on CD20 and CD19 expression. **(E, F)** Comparison of the abundance of the populations identified in (E) between disease states. Significance was assessed using Mann–Whitney U tests and corrected for multiple testing with Benjamini–Hochberg false discovery rate (FDR). **FDR-adjusted *P*-value < 0.01; *FDR-adjusted *P*-value 0.01–0.05.

et al, 2020; Schulte-Schrepping et al, 2020b), we demonstrate that disease progression is dominated by the progressive loss of circulating lymphocytes and gain of myeloid cells. We also detected selective expansion of NK populations and MDSCs, suggesting that

the innate compartment may contribute to the immunological disarray of COVID-19 patients. We then harnessed this multidimensional dataset to generate a machine-learning classifier that could predict disease severity using a defined flow cytometric

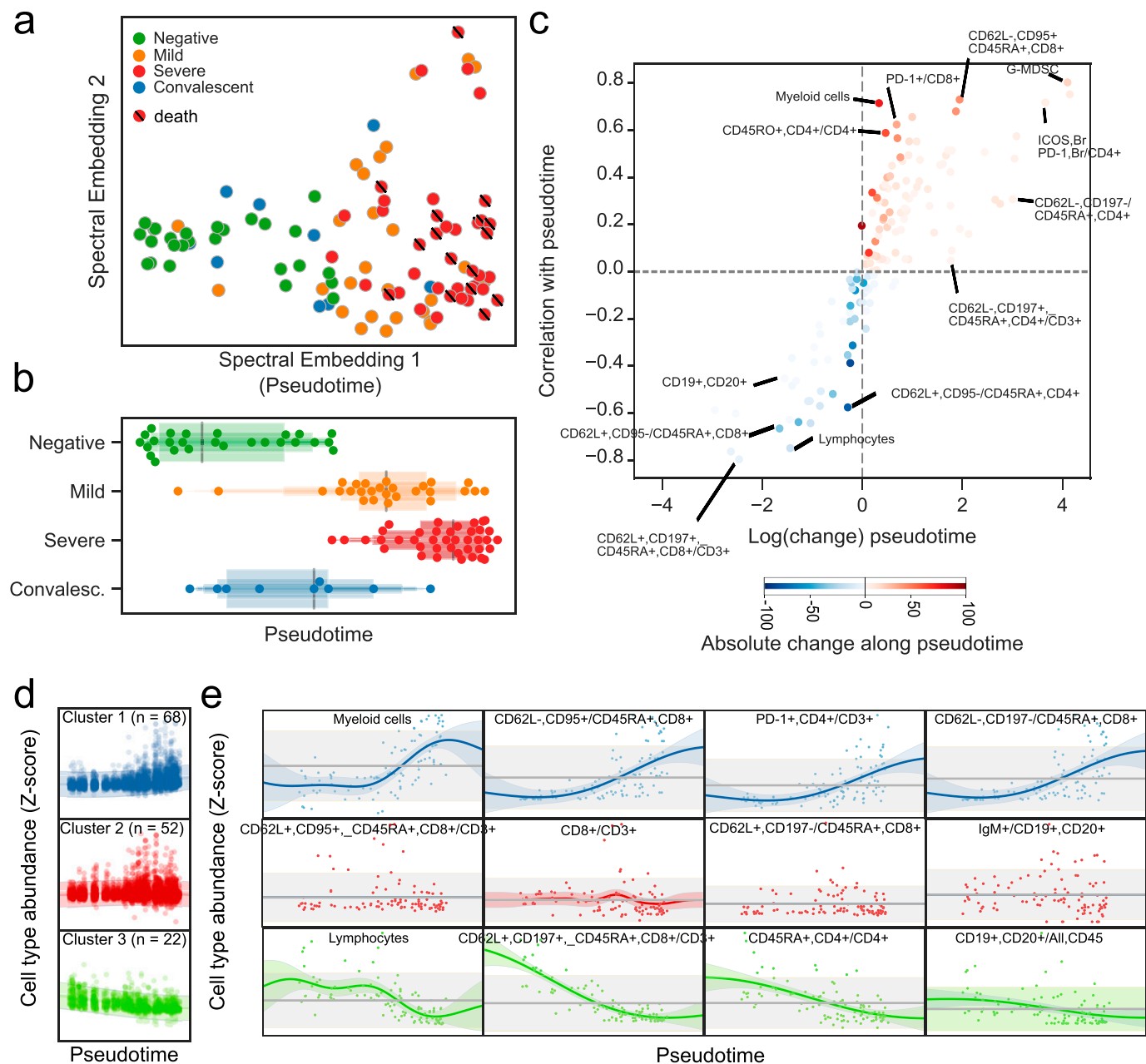

**Figure 5. Pseudo-temporal reconstitution of disease progression reveals a hierarchy of immune changes in COVID-19 disease.**
**(A)** Projection of immune profiles into a two-dimensional latent space that reconstructs the hierarchy of disease progression. The x-axis represents disease progression in the pseudo-temporal space. Sample from patients which died from COVID-19 are marked with a diagonal black line. **(B)** Distribution of samples grouped by disease state along the pseudo-temporal axis derived in (B). **(C)** Immune populations associated with the pseudo-temporal axis represented by either the absolute change in percentage in their extremes (x-axis) or strength of linear association (y-axis). **(D)** Clusters of immune populations based on their abundance along the pseudo-temporal axis. **(D, E)** Examples of immune populations from each cluster in (D).

signature. Our work provides a proof-of-concept that an immune-monitoring algorithm could provide a rapid and personalized approach to manage COVID-19.

Although previous studies have focused on lymphocyte populations (Bellesi et al, 2020; Hadjadj et al, 2020; Kuri-Cervantes et al, 2020; Mathew et al, 2020), to our knowledge the role of innate immune cells is less understood (Agrati et al, 2020). Our study highlights the expansion of MDSCs, especially G-MDSCs, in severe

COVID-19 patients. Unlike their natural counterparts, these elements have suppressive function (Zhou et al, 2018) that impairs immune responses in cancer (Kumar et al, 2016) and derails effective responses against bacterial and viral infections by the adaptive immune system (Bohorquez et al, 2019; Ruan et al, 2020). Given the overall depletion of the immune system's lymphoid branch during COVID-19, an interesting hypothesis is that G-MDSCs and other myeloid cells represent uncontrolled negative feedback.

**Figure 6. Factors conditioning the immune response during COVID-19 and predicting disease severity.**
**(A)** Directed graph of clinical factors (green) and immune populations (pink). Edges represent the association between factors and immune populations and are colored by the direction and strength of association (blue, negative; red, positive). **(B)** Abundance of select immune populations with significantly different responses between sexes dependent on outcome. **(C)** Estimated coefficients of change for severe versus mild disease (left) or tocilizumab treatment (right) for immune populations that change discordantly. **(D, E)** Abundance of select immune populations with significantly different responses between sexes dependent on tocilizumab treatment (D) or intubation (E). **(F)** Graphical depiction of the machine-learning framework for predicting disease severity using the earliest available samples per patient and cross

These elements ultimately contribute to the establishment of pan-immunosuppression, leading to dysregulated responses from the adaptive immune system. It will be essential to establish whether they are actively recruited to infected lungs and whether they are causally involved in disease pathogenesis or represent a systemic compensatory response to inflammation. Because MDSCs are virtually absent in healthy individuals, questions arise regarding the mechanisms of their genesis and tissue recruitment and how they interact with lung tissue. At the same time, our novel observation that NK cells expressing the CD158i variant are over-represented in patients with severe disease raises the question of whether this variant and other KIRs implicate NK cells in disease progression (Fig 3).

Within the adaptive immune system, the mechanisms leading to severe immune depletion, a landmark seen with disease progression and markedly apparent in autopsy samples, are unknown (Bradley et al, 2020). To this end, we observed increased CD25$^+$ T cells in COVID-19 patients, indicating a higher state of activation (Fig 2), but also an increase in CD95$^+$ with disease progression. This phenotype was significantly marked in severe patients, consistent with a recent study (Bellesi et al, 2020). FAS has a crucial role in mediating cell death via FAS ligand engagement, as in activation-induced cell death, or by shifting cells to a more apoptotic-prone phenotype. Although FAS is a natural regulatory checkpoint of T cells, it plays a role in autoimmunity (Suda & Nagata, 1997) and cancer (Chen et al, 2010), and activation-induced cell death is involved in loss of CD4$^+$ and CD8$^+$ cells in HIV patients (Dockrell et al, 1999). However, severely exhausted T cells can undergo apoptosis, and virus-specific T-cell decline can favor viral escape (Moskophidis et al, 1993; Wherry & Ahmed, 2004; Williams & Bevan, 2007). Indeed, similar to previous reports (De Biasi et al, 2020; Zheng et al, 2020a, 2020b), T cells displayed an overall exhausted phenotype, with overexpression of VISTA, TIM3, LAG3, TIGIT, and PD-1 co-inhibitory receptors in COVID-19 patient T cell populations. This likely results in inability of the adaptive immune system to keep viral proliferation in check. In the B-cell compartment, we observed lower expression of CD19 in COVID-19 patients and higher expression of membrane-bound IgM and IgG in both mild and severe patients. These data suggest that under viral exposure, B cells undergo plasmacytoid maturation and immunoglobulin switching. Remarkably, several patients displayed higher IgM than IgG CD19$^+$ CD20$^{+/−}$ cells, suggesting abnormal and delayed maturation of plasma cells (Fig 4). Although the implications remain speculative, they do warrant further investigation given the central role of B cells in the development of immunity by COVID-19 patients.

Taking advantage of our dataset's high-dimensional characteristics and pseudo-temporal modeling, we constructed a COVID-19 disease course landscape. This strategy reveals a continuum of disease progression between healthy state, mild disease, and severe disease. Remarkably, convalescent patients displayed immune phenotypes similar to healthy donors, suggesting a possible return to a largely healthy state, as previously suggested based on the exhaustion phenotype in adaptive responses (Zheng et al, 2020a, 2020b). Conversely, we could speculate that the immune landscape of mild/convalescent patients never achieved the level of disarray observed in severe patients. Although there were marked differences between patients with prevalent mild or severe disease, their recognition remains a unique challenge. One interesting open question is whether the changes associated with mild versus severe disease protect against disease progression or, conversely, which immune populations related to severe illness play a role in the progression to severe disease. Although proof-of-principle, our classifier of severe disease shows robustness and overall value in predicting disease progression based on immune profiling and near-real-time disease monitoring. Thus, it may be valuable to inform clinical action like that proposed in chronic diseases with other high-dimensional assays (Lucas et al, 2020; Unterman et al, 2020 *Preprint*; Zheng et al, 2020a). Moreover, we demonstrated that a classifier with a limited number of markers retains good performance. If confirmed in large cohorts, it could provide a useful approach to stratify patients and predict clinical evolution using a rapid and economical assay.

Although our study confirmed some findings and provides new data on the innate immune landscape of COVID-19 patients, we recognize several limitations. The relatively small sample size and the fact we as others (Juno et al, 2020; Kuri-Cervantes et al, 2020; Mathew et al, 2020) used a control population of healthy individuals that are not age-matched are important drawbacks. Although fully extrapolating our results to a population level should carried out with caution, by comparing the immune profile of younger healthy donors to data from healthy older individuals (n = 5) (Fig S6A), we observed that the main immune changes associated with COVID-19 in our study are still significant (Fig S6B). The lack of standardized time point collection across patients means that temporal inferences across patients may not represent the real disease trajectory of any particular patient and may represent another limitation of our study. Overall, these limitations could be overcome with the support of large population studies that may be better powered to relate new immune populations with disease progression or clinical factors. In this current pandemic, profiling a larger sample of the population and investigating multiple time points systematically may help identify viral adaptation to the host (particularly when coupled to analysis of viral sequence) in patients with different outcomes. These programs may be achieved if an effective institutional organization, multicentric networking, and substantial financial support are available.

The targeted nature of flow cytometry interrogates limited sets of immune populations and implies that only certain molecules can be effectively profiled. In our study we used mainly proportional data when comparing the abundance of immune populations between patient groups. While this may not necessarily imply absolute changes in cell numbers, we observed good overall agreement between changes in proportions and absolute counts when comparing severe and mild disease status (Fig S7). This

---

validation. **(G, H)** Performance of classifiers trained with real or randomly shuffled labels and either all immune populations (G) or with selection for the top most predictive eight populations (H). **(I)** Predicted severity scores over time since symptoms started for immune profiles from patients with at least three longitudinal sampling points. **(G, J)** Relative expression of CD25, CD45RA, and CD45RO over time in four patients from (G). **False discovery rate-adjusted *P*-value < 0.01; *false discovery rate-adjusted *P*-value 0.01–0.05.

highlights the importance of studies using orthogonal modalities such as cytokine profiling (Lucas et al, 2020), single-cell RNA sequencing (Unterman et al, 2020 Preprint; Wilk et al, 2020; Lee et al, 2020; Zhu et al, 2020; Guo et al, 2020a Preprint), and their integration (Su et al, 2020 Preprint; Schulte-Schrepping et al, 2020a Preprint). Nevertheless, even without orthogonal studies, our machine learning approach for predicting disease severity demonstrates predictive potential, although it should be tested in a validation cohort before use in a clinical setting.

Last, we wish to note the work of others and their complementary findings. For example, Laing et al (2020) used peripheral blood flow cytometry and circulating cytokine measurements to demonstrate apparent immune dysregulation in COVID-19 patients. They highlighted additional interesting, complementary features, including increased IL-6, IL-10, and IP-10 and depletions of basophils, plasmacytoid dendritic cells, $T_H1$ cells, and $T_H17$ cells. Incorporating their most differentiating markers with ours could yield a more complete yet targeted panel of markers with more predictive power to determine which patients will rapidly progress to a severe disease state. This incorporation of additional differentiating markers should be pursued in future studies.

Collectively, our study highlights a profound imbalance in the COVID-19 immune landscape, characterized by G-MDSC expansion and T cell exhaustion that may open avenues for clinical translation. Further, our approach provides a powerful tool to predict clinical outcomes and tailor more effective and proactive therapies to COVID-19 patients.

# Materials and Methods

### Study design, sample acquisition, and clinical data

The study was approved by the Institutional Review Board of Weill Cornell Medicine. Participants were recruited from patients hospitalized at New York Presbyterian Hospital from April to July 2020. Some participants in a COVID-19 convalescent plasma donor screening program with prior confirmed diagnosis (by RT-PCR or serology) were given the option to contribute a sample for this research. Acute respiratory distress syndrome was categorized in accordance with the Berlin definition reflecting each subject's worst oxygenation level and with physicians adjudicating chest radiographs (ARDS Definition Task Force et al, 2012). Informed consent was obtained from all participants.

### Flow cytometry

For flow cytometric analysis of circulating leukocytes, peripheral blood was collected in Na-heparin. Except for the MDSC panel, in which PBMCs were prepared by density gradient centrifugation, erythrocytes were lysed with BD Pharm Lyse. Peripheral blood was washed in Dulbecco's PBS (DPBS), lysed in 1× BD Pharm Lyse, and washed again in DPBS. PBMC cell suspensions were prepared with Ficoll-Paque following the manufacturer's protocol. Cells were stored briefly in storage medium (10% heat-inactivated fetal bovine serum/1% L-glutamine/1% pen-strep) before staining with antibody cocktails.

For each panel, one million cells were stained with specific cocktails of fluorochrome-conjugated antibodies (Tables S3 and S4). Cells were washed with DPBS and then stained with dead cell dye (BD Fixable

Viability Stain 700) before washing with wash buffer (0.5% BSA/DPBS/NaN₃). Cells were then treated with 50 $\mu$l of Fc-blocking solution (2% normal rabbit serum/10% BD Fc Block/DPBS) before application of a 100-$\mu$l antibody cocktail diluted in wash buffer. Samples were stained within 6 h of sample collection and analyzed on a BD Biosciences FACSCanto flow cytometer within 2 h of staining. The stopping gate was set to acquire 500,000 viable, nucleated single cells.

### Supervised quantification of immune cell populations (gating)

Immune populations were quantified by manual analysis with BD FACSDiva. Absolute counts of populations were exported to comma separated values and relative population sizes were calculated in Microsoft Excel. Gating for each panel started with a time gate, followed by a singlet gate (FSC-A versus FSC-H). Next, viable cells (dead cell dye versus FSC-A) and nucleated cells (FSC-A versus SSC-A) were gated. Populations of MDSCs were gated sequentially from leukocytes (CD45 versus SSC-A), then CD3/CD56/CD19 (Lin)- and HLA-DR$^{-/dim}$ cells, followed by CD33$^+$ and CD11b$^+$ cells. From there, granulocytic cells were defined as CD14$^-$ and CD15$^+$, monocytic cells as CD14$^+$ and CD15$^{-/dim}$, and immature forms as CD14$^-$ CD15$^{-/dim}$.

$T_{REG}$ were defined by sequential gating of lymphocytes (CD45 versus SSC), T cells (CD45 versus CD3), T helper (CD8 versus CD4), and finally $T_{REG}$ were defined as CD127 dim and CD25$^+$. The $T_{FH}$ panel was gated the same as in the T cell regulatory panel down to the CD4$^+$ helper gate. Under this gate, CD185+ cells were quantified (CD185 versus CD8) and the ICOS bright, PD-1 bright (CD278 versus CD279) cells were gated. The ICOS bright, PD-1 bright $T_{FH}$ gate was placed under the CD185+ gate to identify the population with all the phenotypic markers of $T_{FH}$ lineage in this panel.

The analysis of the T cell memory and checkpoint panels started with identifying T cells (CD3 versus SSC), then the CD4 helper and CD8 cytotoxic subsets. To analyze the T-cell checkpoint panel, individual exhaustion markers were gated on histogram plots. The T-cell memory panel was further subdivided into CD45RA$^+$/CD45RO$^-$ and CD45RA$^-$/CD45RO$^+$ subsets. Under these gates, two quadrant gates were placed on CD62L versus CCR7 and CD62L versus FAS.

Gating for the B-cell panel began with CD45 versus CD19 then FSC-A and SSC-A to identify cells of the B lineage. The CD20$^+$ and CD20$^-$ subsets were gated (CD19 versus CD20) and IgG and IgM were quantified within the CD20$^+$ subset (IgM versus IgG).

NK cells were identified by sequential gates on CD56 versus CD3, FSC-A versus SSC-A, and CD56 versus CD16. CD56$^+$, CD16 bright, mature NK cells were then interrogated for their reactivity with individual anti-KIR (CD158) and anti-NKG2A (CD159a) antibodies with gates on histogram plots. KIR-negative NK were identified by sequential gating on CD158a versus CD158b double-negative, then CD158i and CD158e double-negative subsets.

### Statistical testing

Nonparametric Mann–Whitney U tests were used to assess the significance of pairwise changes in the proportions of immune populations between severity groups using the Pingouin package, version 0.3.7. Multiple test correction was performed with the Benjamini–Hochberg FDR method.

### Single-cell analysis of immune cell populations

To select cells from the events, single cells were gated using forward-side scatter height and area, CD45-positivity, viability dye-negativity, and the major marker of each panel (e.g., CD3 for T cell memory panel). Compensation was applied using FlowKit (White, 2020) version 0.5.0, and an inverse hyperbolic transformation (AsinhTransform) was applied with parameters t = 10,000, m = 4.5, a = 0. To construct a shared latent representation for all cells, dimensionality reduction was performed with principal component analysis, a neighbor graph was computed using 15 neighbors per cell, UMAP (Becht et al, 2018) with default parameters, and Leiden clustering, all using the Scanpy package (Wolf et al, 2018) version 1.5.1. For each discovered single-cell cluster, a proportion of cells were calculated in relation to a specific clinical factor after normalization by the frequency of the same factor in the cohort.

### Pseudotime inference and time series modeling of immune cell dynamics during disease progression

To learn a latent manifold of the data, the nonlinear method Laplacian Eingenmaps (Belkin & Niyogi, 2003) was used as implemented in the "*SpectralEmbedding*" method of the scikit-learn framework (Pedregosa & Varoquaux, 2011) (version 0.23.0) with default parameters. A z-scored matrix of proportional data was input for all immune cell populations (variables) and patient samples (observations). To rank the features by their association with the learned space, Pearson's correlation was calculated between the first component and each variable, in addition to the fold and absolute change in the variable between the top and bottom 10% of the samples in each extreme of the embedding. The same procedure applied to a Uniform Manifold Approximation and Projection (UMAP) latent representation of the same data yielded similar results, with the exception that the spread of samples according to disease progression was parallel to multiple learned axes rather than single.

To rank variables by the amount of change in both real time since the reported start of symptoms for a single patient or over the learned latent space across all patients, GPy package (GPy) was used to fit Gaussian Process regression models on the learned pseudotime axis (independent variable) and the abundance of each immune cell population (dependent variables). A variable radial basis function kernel and a constant kernel (both with an added noise kernel) were fitted and the log-likelihood and SD of the posterior probability of the two were compared as described previously (Rendeiro et al, 2020). To cluster the abundance of immune populations based on their dynamics over the pseudotime axis, the same kernels were used to fit a Mixture of Hierarchical Gaussian Processes (MOHGP) as implemented in the GPClust package (Hensman et al, 2012 *Preprint*, 2013) using eight as an initial guess of number of clusters.

### Linear modeling of immune cell type abundances

Because of the proportional nature of the dataset, generalized linear models were fit using a $\gamma$-distributed noise model with a log-link function. Ridge regularization was used to ensure robust coefficients given the low abundance of some populations, and the model was fit with ordinary least squares optimization using the

*statsmodels* package (Seabold & Perktold, 2010) version 0.11.1. Categorical variables were one-hot encoded and numeric ones such as age or days since symptoms started were kept as years or days, respectively; the date of acquisition was transformed into days and scaled to the unit interval. Because values for clinical categorical variables and comorbidities were only available to COVID-19 patients, various models were used that aimed to explore different aspects of immune system change during COVID-19:

1. Comparison of healthy donors to COVID-19 patients: sex + race + age + batch + COVID-19.
2. Effect of clinical/demographic factors on COVID-19 patients: sex + race + batch + COVID-19 + severity group + hospitalization + intubation + death + diabetes + obesity + hypertension + age in years + days since symptoms start.
3. Effect of tocilizumab treatment on severe patients only: sex + age + batch + tocilizumab.

To generate a graph of interactions between factors and immune populations, significant coefficients (FDR-adjusted *P*-value < 0.05) were used as undirected edges between factors and immune populations. For edges between factors, the Pearson correlation between factors across immune populations was used. Exclusively for visualization, coefficients for the continuous variables "age" and "time since symptoms started" were multiplied by half of the median of the values of that variable (33.0 and 10.8, respectively) to make the range of coefficients comparable with the categorical variables. Visualizations were produced using Gephi version 0.9.2 with the Force Atlas2 layout with parameters "LinLog mode," "scaling factor" 8.0, and "gravity" 11.0.

### Prediction of disease severity from immunotypes

A Random Forest Classifier was trained as implemented in *scikit-learn* framework (Pedregosa & Varoquaux, 2011) (version 0.23.0) to distinguish between cases with "mild" and "severe" disease using 10-fold cross validation. The cross validation loop was repeated 100 times and models were fit with real or randomized labels. Test set performance was assessed with the ROC AUC. To investigate the performance of the classifier, feature importance was averaged across cross validation folds and iterations and the log fold importance of the real models over the randomized labels was calculated. A sign was added to the feature importance depending on the sign of the Pearson correlation of each variable with each class. Only the earliest temporal sample of each patient was used to ensure lack of data leakage (avoid training/testing on samples from the same patient without stratified cross validation) and to maximize the utility of the model. The same cross validation scheme was used to develop a classifier using a subset of features but including feature selection using mutual information inside the cross validation loop. To predict severity longitudinally for single patients, a model was trained on the initial samples from all other patients and tested on the samples of the patient in question.

## Data Availability

Quantification of immune cell populations is available as a Supplementary Table file. Hierarchical data format files with single cell

data (h5ad) are available as indicated in the repository with source code for the study (https://github.com/ElementoLab/covid-flowcyto).

# Supplementary Information

# Acknowledgements

This project was supported by a Translational Pathology Research COVID-19 grant to G Inghirami and by the National Center for Advancing Translational Science of the National Institute of Health Under Award Number UL1TR002384 to O Elemento and M Salvatore. AF Rendeiro is supported by the National Cancer Institute grant T32CA203702. CK Vorkas is supported by National Institutes of Health (NIH) K08 AI132739; A Morales is supported by grant KL2TR002385 of the Clinical and Translational Science Center at Weill Cornell Medical College. K Saito is supported by NIH K08 AI139360; CD Brown is supported by NIH T32 AI07613-19 (PI: Gulick) and by the Kellen Foundation. L Galluzzi is supported by from the Leukemia and Lymphoma Society (LLS), a startup grant from the Dept. of Radiation Oncology at Weill Cornell Medicine (New York, US), a Rapid Response Grant from the Functional Genomics Initiative (New York, US). We thank Andrew Marderstein, Fayzan Chaudhry, and Liron Yoffe for helpful discussions on the machine learning classifier for disease severity. We are grateful for the support of members of the Immunopathology laboratory at New York Presbyterian Hospital, Weill Cornell Medicine, whose dedication and contribution have been instrumental for the execution of this project. We are grateful to the patients and their family who agreed to be part of the study and all the medical staff who cared for them.

## Author Contributions

AF Rendeiro: conceptualization, data curation, software, formal analysis, visualization, methodology, project administration, and writing—original draft, review, and editing.
J Casano: resources, data curation, formal analysis, investigation, methodology, and writing—review and editing.
CK Vorkas: conceptualization, resources, data curation, supervision, investigation, methodology, and writing—original draft, review, and editing.
H Singh: data curation and writing—review and editing.
A Morales: data curation and investigation.
RA DeSimone: data curation and writing—review and editing.
GB Ellsworth: data curation and writing—review and editing.
R Soave: data curation.
SN Kapadia: data curation.
K Saito: data curation.
CD Brown: data curation.
J Hsu: data curation.
C Kyriakides: data curation, investigation, and writing—review and editing.
S Chiu: resources, data curation, formal analysis, validation, investigation, and methodology.
LV Cappelli: data curation and writing—review and editing.
MT Cacciapuoti: data curation.
W Tam: data curation.
L Galluzzi: conceptualization and writing—original draft, review, and editing.
PD Simonson: data curation, software, formal analysis, investigation, and writing—review and editing.
O Elemento: conceptualization, resources, supervision, funding acquisition, project administration, and writing—original draft, review, and editing.
M Salvatore: conceptualization, resources, data curation, supervision, funding acquisition, investigation, methodology, project administration, and writing—original draft, review, and editing.
G Inghirami: conceptualization, resources, data curation, supervision, funding acquisition, investigation, methodology, project administration, and writing—original draft, review, and editing.

## Conflict of Interest Statement

The authors declare that they have no conflict of interest.

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
