## [Reviewer comments · Life Science Alliance]

Life Science Alliance

Profiling of immune dysfunction in COVID-19 patients allows early prediction of disease progression

Andre Rendeiro, Joseph Casano, Charles Vorkas, Harjot Singh, Ayana Morales, Robert DeSimone, Grant Ellsworth, Rosemary Soave, Shashi Kapadia, Kohta Saito, Christopher Brown, JingMei Hsu, Christopher Kyriakides, Steven Chiu, Luca Vincenzo Cappelli, Maria Cacciapuoti, Wayne Tam, Lorenzo Galluzzi, Paul Simonson, Olivier Elemento, Mirella Salvatore, and Giorgio Inghirami

DOI: <https://doi.org/10.26508/lsa.202000955>

Corresponding author(s): *Giorgio Inghirami, Weill Cornell Medicine and Mirella Salvatore, Weill Cornell Medicine*

Review Timeline:	Submission Date:	2020-11-12
	Editorial Decision:	2020-11-16
	Revision Received:	2020-11-25
	Editorial Decision:	2020-12-01
	Revision Received:	2020-12-03
	Accepted:	2020-12-04

Scientific Editor: Shachi Bhatt

Transaction Report:

Please note that the manuscript was previously reviewed at another journal and the reports were taken into account in the decision-making process at Life Science Alliance. Since the original reviews are not subject to Life Science Alliance's transparent review process policy, the reports and author response cannot be published.

November 16, 2020

Re: Life Science Alliance manuscript #LSA-2020-00955-T

Dr. Andre F Rendeiro
Weill Cornell Medicine
New York

Dear Dr. Rendeiro,

Thank you for transferring your manuscript entitled "Longitudinal immune profiling of mild and severe COVID-19 reveals immune dysfunction and allows prediction of clinical progression" to Life Science Alliance (LSA).

For a brief overview, the manuscript was reviewed at one of our sister journals, where it was rejected due to lack of advance over another recently published manuscript. As per LSA's mission, we do not consider lack of conceptual advance to be a hindrance to publication of a well done study. LSA editors reviewed the manuscript and the accompanying reviewers' comments, and determined that the study is publishable at LSA pending following minor revisions,

- + Respond to R1's concern about the use of younger healthy donors as controls with discussion drawing the readers' attention to this caveat and a discussion on how this impacts the conclusions drawn (R1 pt1)
- + Clarify the need to stratify severe COVID-19 patients as incubated vs not (R1 pt 3)
- + Improve data presentation (R2, main point and pt1)
- + Tone down the focus on longitudinal aspect of the data as pointed out by R2 (pt 2)
- + Improve designation of MDSCs in patients and discuss and compare the recent Schultze-Schrepping et al Cell manuscript (R2 pt 3)

Thank you for this interesting contribution to Life Science Alliance. We are looking forward to receiving your revised manuscript.

Sincerely,

Shachi Bhatt, Ph.D.
Executive Editor
Life Science Alliance
<https://www.lsjournal.org/>
Tweet @SciBhatt @LSAJournal

- A letter addressing the reviewers' comments point by point.
- An editable version of the final text (.DOC or .DOCX) is needed for copyediting (no PDFs).
- High-resolution figure, supplementary figure and video files uploaded as individual files: See our detailed guidelines for preparing your production-ready images, <https://www.life-science-alliance.org/authors>
- Summary blurb (enter in submission system): A short text summarizing in a single sentence the study (max. 200 characters including spaces). This text is used in conjunction with the titles of papers, hence should be informative and complementary to the title and running title. It should describe the context and significance of the findings for a general readership; it should be written in the present tense and refer to the work in the third person. Author names should not be mentioned.

B. MANUSCRIPT ORGANIZATION AND FORMATTING:

December 1, 2020

RE: Life Science Alliance Manuscript #LSA-2020-00955-TR

Giorgio Inghirami
Weill Cornell Medicine

Dear Dr. Inghirami,

Thank you for submitting your revised manuscript entitled "Profiling of immune dysfunction in COVID-19 patients allows early prediction of disease progression". We would be happy to publish your paper in Life Science Alliance pending final revisions necessary to meet our formatting guidelines.

Along with the points listed below, please also attend to the following,

- please upload your supplementary tables as single editable doc or excel files
- please add a callout for Figure 3E, Figure S6A,B, and Figure S7A-E in your main manuscript text
- please use the [10 author names, et al.] format in your references (i.e. limit the author names to the first 10)

A. FINAL FILES:

B. MANUSCRIPT ORGANIZATION AND FORMATTING:

Sincerely,

Shachi Bhatt, Ph.D.
Executive Editor
Life Science Alliance
<https://www.lsjournal.org/>
Tweet @SciBhatt @LSAJournal

December 4, 2020

RE: Life Science Alliance Manuscript #LSA-2020-00955-TRR

Giorgio Inghirami
Weill Cornell Medicine
Pathology and Laboratory Medicine
1300 York Avenue
New York 10065

Dear Dr. Inghirami,

Thank you for submitting your Research Article entitled "Profiling of immune dysfunction in COVID-19 patients allows early prediction of disease progression". It is a pleasure to let you know that your manuscript is now accepted for publication in Life Science Alliance. Congratulations on this interesting work.

DISTRIBUTION OF MATERIALS:

Again, congratulations on a very nice paper. I hope you found the review process to be constructive and are pleased with how the manuscript was handled editorially. We look forward to future exciting submissions from your lab.

Sincerely,

Shachi Bhatt, Ph.D.

Executive Editor

Life Science Alliance

<https://www.lsjournal.org/>
